# WGS of Commensal *Neisseria* Reveals Acquisition of a New Ribosomal Protection Protein (MsrD) as a Possible Explanation for High Level Azithromycin Resistance in Belgium

**DOI:** 10.3390/pathogens10030384

**Published:** 2021-03-23

**Authors:** Tessa de Block, Jolein Gyonne Elise Laumen, Christophe Van Dijck, Said Abdellati, Irith De Baetselier, Sheeba Santhini Manoharan-Basil, Dorien Van den Bossche, Chris Kenyon

**Affiliations:** 1Department of Clinical Sciences, Institute of Tropical Medicine, 2000 Antwerp, Belgium; tdeblock@itg.be (T.d.B.); jlaumen@itg.be (J.G.E.L.); cvandijck@itg.be (C.V.D.); sabdellati@itg.be (S.A.); idebaetselier@itg.be (I.D.B.); sbasil@itg.be (S.S.M.-B.); dvandenbossche@itg.be (D.V.d.B.); 2Laboratory of Medical Microbiology, Vaccine and Infectious Disease Institute, University of Antwerp, 2000 Antwerp, Belgium; 3Department of Medicine, University of Cape Town, Cape Town 7701, South Africa

**Keywords:** antimicrobial resistance, commensal *Neisseria*, horizontal gene transfer, *msrD*

## Abstract

In this study, we characterized all oropharyngeal and anorectal isolates of *Neisseria* spp. in a cohort of men who have sex with men. This resulted in a panel of pathogenic *Neisseria* (*N. gonorrhoeae* [n = 5] and *N. meningitidis* [n = 5]) and nonpathogenic *Neisseria* (*N. subflava* [n = 11], *N. mucosa* [n = 3] and *N. oralis* [n = 2]). A high proportion of strains in this panel were resistant to azithromycin (18/26) and ceftriaxone (3/26). Whole genome sequencing (WGS) of these strains identified numerous mutations that are known to confer reduced susceptibility to azithromycin and ceftriaxone in *N. gonorrhoeae*. The presence or absence of these known mutations did not explain the high level resistance to azithromycin (>256 mg/L) in the nonpathogenic isolates (8/16). After screening for antimicrobial resistance (AMR) genes, we found a ribosomal protection protein, Msr(D), in these highly azithromycin resistant nonpathogenic strains. The complete integration site originated from *Streptococcus pneumoniae* and is associated with high level resistance to azithromycin in many other bacterial species. This novel AMR resistance mechanism to azithromycin in nonpathogenic *Neisseria* could be a public health concern if it were to be transmitted to pathogenic *Neisseria*. This study demonstrates the utility of WGS-based surveillance of nonpathogenic *Neisseria*.

## 1. Introduction

*Neisseria gonorrhoeae* has developed antimicrobial resistance (AMR) to every class of antimicrobials used to treat it [1]. Of particular concern are the rapid increases in azithromycin resistance and the emergence of combined ceftriaxone and high-level azithromycin resistance [1,2,3,4]. Much of these resistance mechanisms are acquired via horizontal gene transfer (HGT) from commensal *Neisseria* [5,6,7,8]. The importance of this pathway for the emergence of AMR has been well established for extended spectrum cephalosporins, such as ceftriaxone. Phylogenetic analyses confirmed that the HGT of a section of the *penA* gene from commensal *Neisseria* played a crucial role in the genesis of ceftriaxone resistance in *N. gonorrhoeae* [7,9]. As a result, one of the crucial first steps in the emergence of ceftriaxone resistance in *N. gonorrhoeae* was the selection of ceftriaxone resistance in commensal *Neisseria* [7,9,10,11]. Similarly, HGT played an important role in the genesis of AMR to macrolides such as azithromycin [8].

An important reason why AMR would be expected to emerge in commensal *Neisseria* prior to *N. gonorrhoeae* is the considerably higher prevalence of commensals in human populations. Whilst the prevalence of *N. gonorrhoeae* is typically a fraction of a percent in general populations and only reaches 10% in core groups [1,12], the prevalence of commensal *Neisseria* is close to 100% in the oropharynx, where they form an important part of a healthy microbiome [6,13]. This makes them more likely to develop AMR in response to high antimicrobial consumption compared to *N. gonorrhoeae* [10,14]. Because this AMR can be transferred to *N. gonorrhoeae* and monitoring AMR in commensals is not complicated, it has been proposed that the surveillance of AMR in commensals could be used as an early warning system to detect the risk of AMR in *N. gonorrhoeae* and other bacteria [10,14,15].

AMR in *N. gonorrhoeae, Mycoplasma genitalium* and *Treponema pallidum* has frequently emerged in core-groups with high rates of partner turnover and high antimicrobial consumption [14,16,17,18]. This provides the rationale for monitoring the antimicrobial susceptibility of commensal *Neisseria* in these core-groups. In our setting, men who have sex with men (MSM) attending our STI clinic represent one such core-group [17,19]. In a pilot study, we assessed ceftriaxone and azithromycin susceptibilities of all *Neisseria* spp. isolated (n = 26) pre-and post-treatment from 10 MSM attending our STI clinic with a diagnosis of anorectal gonorrhea. The minimum inhibitory concentrations (MICs) were found to be alarmingly high [15]. The most prevalent commensal, *N. subflava*, for example, was found to have a median azithromycin MIC of 176 mg/L (IQR 0.047–256) and a median ceftriaxone MIC of 0.38 mg/L (IQR 0.023-2) [15]. These values were considerably higher compared to Belgian historical samples and moreover, were the highest MIC values for *N. subflava* ever published [5,14,15,20].

In this paper, we reported the antimicrobial resistance-associated mutations detected in these *Neisseria* species with a focus on those implicated in macrolide and extended-spectrum cephalosporin resistance.

## 2. Results

### 2.1. Characterization of Strain Collection

Twenty-six clinical Neisseria species were isolated and subjected to whole genome sequencing. De novo assemblies resulted in genome sizes ranging from 2.09 Mbp to 2.82 Mbp with the number of putative coding DNA sequence (CDS) ranging from 1983 to 2535 (Appendix A).

Prior species identification by matrix-assisted laser desorption/ionization—time of flight mass spectrometry (MALDI-TOF–MS) was verified by analyzing the whole genome assemblies using BIGSdb and the 50S ribosomal gene, *rplF* [15,21,22]. There was complete agreement between these three methods except that MALDI-TOF misidentified *N. mucosa* as *N. macacae,* and *rplF* gene annotation could not discriminate between *N. mucosa* and *N. oralis*. In the present study, screening by BIGSdb was therefore used in species identification. This resulted in a *Neisseria* panel with a composition of pathogenic *Neisseria*: *N. gonorrhoeae* (n = 5) and *N. meningitidis* (n = 5) and nonpathogenic *Neisseria*: *N. subflava* (n = 11), *N. mucosa* (n = 3) and *N. oralis* (n = 2).

The antimicrobial susceptibility testing (AST) of this panel showed higher azithromycin MICs in commensal Neisseria isolates than in pathogenic Neisseria (commensal MIC median 140 mg/L (IQR 6-256), pathogens median 0.75 mg/L (IQR 0.31-1.00; *p*-value < 0.0001; Table 1; Appendix A). The same was true for ceftriaxone (commensal MIC median 0.082 mg/L (IQR 0.043–0.125), pathogens median 0.016 mg/L (IQR 0.016–0.016; *p* -value < 0.0001; Table 1; Appendix A). For illustrative purposes, we applied The European Committee on Antimicrobial Susceptibility Testing (EUCAST) definitions of gonococcal resistance to azithromycin and ceftriaxone to define AMR for all the Neisseria spp. [23] (Figure 1). Of note, all the commensal Neisseria were classified as resistant to azithromycin (MIC > 1 mg/L). Resistance was particularly evident in the N. subflava strains, of which 8/11 had azithromycin MICs greater than 256 mg/L. Three of these eight isolates were classified as resistant to ceftriaxone.

The combined MICs of all post-treatment isolates (visit 2) where higher than those of the pre-treatment isolates (visit 1): the azithromycin MIC increased from a median of 1.5 mg/L (IQR 0.016–0.047) to 256 mg/L (IQR 24–256; *p*-value 0.0012) and ceftriaxone MIC from 0.016 mg/L (IQR 0.016-0.047) to 0.094 mg/L (IQR 0.047–0.125; *p*-value 0.0059). This effect was mainly driven by the elimination of the more susceptible pathogenic Neisseria between visits 1 (n = 8) and 2 (n = 0; Figure 2). There was no statistically significant increase in azithromycin or ceftriaxone MIC obtained from visit 2 compared to visit 1 in the N. subflava isolates alone. The sample sizes were too small to evaluate MIC changes within other species.

### 2.2. SNP Determination

We align genes that are known to mediate macrolide and cephalosporin resistance in *N. gonorrhoeae* (*mtrD*, *mtrR*, *penA*, *ponA*, *rplD*, *pilQ* and 23S rRNA). In so doing, we found 14 single nucleotide polymorphism (SNPs) in commensal *Neisseria*, which are well established resistance-associated mutations (RAMs) to macrolides/cephalosporins (Figure 3) [24,25]. Five of these were present in at least half the isolates: penA I312M (16/16), penA V316T (16/16), penA A517G (10/16) and mtrD K823E/S (16/16). Less prevalent SNPs detected were: penA A311V (3/16), penA T484S (1/16), penA A501V (1/16), penA N513Y (1/16), penA G543S (2/16), penA G546S (2/16), penA P552L (1/16), rplD G70S (1/16) and rplD G70A (1/16).

In the pathogenic Neisseria, the following 10 RAMs were found: mtrD K823E/S (3/10), mtrR A39T (3/10), mtr promoter (3/10), penA I312M (1/10), penA V316T (1/10), penA A51V (1/10), penA N513Y (1/10), penA A517G (5/10), porB G120K/A121N (1/10) and porB Δ121 (5/10; Figure 3).

No RAMs in PilQ or 23S rRNA were found in either the commensal or pathogenic *Neisseria*.

The identification of RAMs in regions which are more divergent among species, like *porB* loop III and *mtr* promoter regions, are challenging to interpret.

***porB:*** In the commensals, up to four copies of full length *porB* were detected per isolate. In *N. subflava,* four paralogs were detected. These varied in size between 997 to 1198 bp. In *N. mucosa,* three paralogs were found (length of 984 bp, 1074 bp and 1122 bp), whereas in *N. oralis,* two paralogs of 1098 bp and 1125 bp were present. Mutations in the loop III region in *porB*, known as *penB*, are associated with reduced susceptibility to antimicrobials such as ceftriaxone [24,25]. Whilst the transmembrane domains of *porB* genes in commensals are relatively conserved, the loop III structure is hypervariable among *Neisseria* species and therefore, *penB* mutations are difficult to interpret in commensals (Appendix A).

***mtrCDE:*** Alignment of the *mtr* promoter showed variable and conserved regions between the *Neisseria* species (Figure 4). More specifically, no differences were found in the -10 *mtrCDE* and -10 *mtrR* promoter sites, but there were variations in the *mtrR* binding site and in the inverted repeat. An A to C conversion in the inverted repeat of the -35 *mtrCDE* promoter was detected in *N. gonorrhoeae* (1/5). This A to C conversion was also present in *N. meningiditis* (5/5), *N.oralis* (2/2) and *N. mucosa* (3/3). All strains of *N. subflava* (10/10) contained a G at this position. One strain of *N. meningitidis* (MO00021/1) contained a single T insertion in the inverted repeat region, which has previously been described in a clinical *N. gonorrhoeae* strain [26]. One *N. gonorrhoeae* strain (MO00014/1, MIC of 2 mg/L for azithromycin) contained the A to C conversion in the inverted repeat region as well as a G to A conversion in the -35 *mtrR* promoter site. This G to A conversion was also detected in 2 *N. subflava* strains (MO00036/1 and MO00036/2). A commonly described disruption of the -35 *mtrCDE* promoter, deletion of one A in the inverted repeat was not found in this panel [27].

### 2.3. AMR Gene Screening

In 13 commensals and in one pathogenic *Neisseria* isolate, gene acquisition was detected (Figure 5). In total, six different genes related to resistance to five different classes of antimicrobials were found. Only one of these is implicated in resistance to macrolides -*msr(D).*

***Msr(D)*:** The *msr(D)* gene has been shown to confer macrolide resistance in numerous species by acting as a ribosomal protection protein and displacing the macrolides from the ribosome [28,29,30]. It has not been reported in *Neisseria* before. However, it was present in 9 out of 11 isolates of *N.subflava* in our panel. Its genomic location and organization were highly conserved in all nine *N. subflava* isolates (Appendix A). A 1944 bp sequence was present at 30 bp downstream of a DNA uptake sequence (DUS) and continued 457 bp downstream and 23 bp upstream of *msr(D)* (Figure 6). This entire 1944 bp sequence was 100% identical to a portion (position 2006 to 3950) of the macrolide efflux genetic assembly (MEGA) element in *Streptococcus pneumoniae* [31,32,33]. This element encodes five open reading frames (ORFs); *mef(A)*, *msr(D)*, ORF6, ORF7 and *umuC-mucB*. The integrated portion in *N. subflava* includes the complete *msr(D)* gene and a 338 bp section from the C terminus of *mef(A)*. The *msr(D)* gene is part of the antibiotic resistance ATP Binding Cassette (ARE ABC-F) protein family, which all have a similar basic structure. A unique domain in this family is the P site tRNA Interaction Motif (PtlM) which is thought to play a crucial role in displacing bound macrolides from the 50S ribosome [34]. This motif as well as two other ABC-F unique motifs were found in the *msr(D)* gene (Appendix A). To assess whether *msrD* may be playing a role in azithromycin resistance, genotypic azithromycin MIC prediction models with and without *msrD* were compared. These models revealed that the known azithromycin RAMs were only able to explain 21% of the variation in azithromycin MICs in nonpathogenic *Neisseria* (adjusted R-squared: 0.21, *p*-value: 0.18), compared to 96% in pathogenic *Neisseria* (adjusted R-squared 0.96, *p*-value: 0.001). The inclusion of a binary variable indicating the presence or absence of the *msr(D)* gene improved the predictive ability of the model from 21 to 84% in nonpathogenic *Neisseria* (adjusted R-squared: 0.84, *p*-value: <0.001).

**tetM:** The tetM gene was detected in one N. gonorrhoeae and four N.subflava strains (Figure 5). Whilst the tetM gene was carried on a plasmid in N. gonorrhoeae, in N. subflava it was integrated into the chromosome (Appendix A).

### 2.4. Core Genome (cg) MLST

A gene-by-gene approach was used to assess the cgMLST/wgMLST allelic loci differences. In total, 3799 allelic loci were identified using the study strains (n = 26) and the reference strains (n = 7). For constructing the cgMLST tree, 1038 allelic loci were used (Figure 7). Distinct clusters were observed and the strains clustered closely according to their respective species. Maximum allelic differences between *N. gonorrhoeae* reference strains and the *N. subflava* core genome were >3000. The maximum allelic difference of the two strains which lacked the msr(D) had allelic loci differences of >4000. Interestingly, both strains originated from different clones.

## 3. Discussion

In this small study, we explored the genetic determinants of the high azithromycin and ceftriaxone MICs identified in *Neisseria* isolates from an exploratory study among Belgian MSM. The ceftriaxone MICs could be fairly accurately predicted by known RAMs in *penA*. The high prevalence of these RAMs in *N. subflava* is a cause of concern as the HGT of these RAMs from commensal *Neisseria* has been shown to be responsible for the emergence of cephalosporin resistance in the past [9]. Although known macrolide RAMs were identified in commensals, these did not fully explain the high azithromycin MICs values in these commensals. In particular, none of the rRNA 23S RAMs known to cause high level azithromycin resistance in *N. gonorrhoeae* were found [26]. The addition of *msr(D)* to the regression models improved the model prediction from 21 to 84%.

An important limitation of this study is that we did not experimentally validate the effect of the *msr(D)* gene on macrolide MICs. Furthermore, we did not evaluate whether this gene could be transformed into *N. gonorrhoeae* or *N. meningitidis*. The ability of the ABC-F proteins, including *msr(D)*, to induce macrolide resistance in a range of Gram-positive and -negative bacterial species is however well established [34,35,36,37,38]. We demonstrated that the complete *msr(D)* gene has been incorporated into the chromosomal DNA of nine of 11 circulating *N. subflava* strains. These nine isolates exhibit 100% sequence identity in this gene and the surrounding DNA to the MEGA element in *S. pneumoniae* [31]. These findings suggest that the *msr(D)* gene may have been acquired from *S. pneumoniae* (or other bacterial species [35,36]) via transformation or less likely a plasmid. The presence of a DUS just upstream from the integration site increases the probability that transformation was responsible [39]. Core genome MLST analysis suggests that these acquisitions in *N. subflava* either took place on more than one occasion or that the *msr(D)* has been taken up and lost in sub-lineages.

In *S. pneumoniae,* the MEGA element includes the full length *mef(A)* efflux pump, which is thought to act synergistically with msr(D). Msr(D) displaces macrolides from ribosomes and the mef(A) then expels the macrolides before they can reattach [28,36]. The truncated version of mef(A) in *N. subflava* is unlikely to be able to perform this efflux function. It is possible that other efflux pumps perform this function. Of note, all 11 of the *N. subflava’s* in this study contained the K823E mutation in *mtrD* which has been shown to enhance the ability of the *mtrCDE* efflux pump to export macrolides [8]. The retained portion of the MEGA element contains the putative upstream regulatory MYLIFM sequence [34]. Various lines of evidence suggest that this sequence could enable the *msr(D)* gene to be induced by macrolides [29,35]. Direct experimental evidence for this effect is however only available for *msr(A)* and *vmlA* in the ABC-F gene family [28,34].

We also found evidence that another gene that confers antimicrobial resistance via interactions with the ribosome, *tetM*, has been taken up by a commensal *Neisseria* and integrated into its chromosome. This uptake and integration was originally described in 1987, where once again a Streptococcal species was thought to be the donor [40,41,42]. More recently, Fiore et al. have found *tetM* to be present in *N. subflava.* They did not, however, specify whether it was carried on a plasmid or integrated into the chromosome [5].

In addition to the acquisition of the *msr(D*) and *tetM* genes, several SNPs in commensals were identified which could be a source of RAMs in *N. gonorrhoeae* and *N. meningitidis*. In keeping with previous studies, we found that the mtr promoter region, *penA*, *mtrD* in commensals are a potential rich source of RAMs [7,8,9]. Whilst the G70S and G70A mutations in *rplD* have been shown to lead to reduced susceptibility to macrolides in *N. gonorrhoeae*, their presence in commensal *Neisseria* has not, to the best of our knowledge, been established before [43].

The high ceftriaxone and azithromycin MICs in our circulating *Neisseria* are a cause of concern. They are in all likelihood a result of the high antimicrobial consumption of the surveyed population [10,15]. The consumption of macrolides and cephalosporins is high in the general Belgian population compared to other European countries [44,45]. This has been linked to a combination of cultural factors (such as a high uncertainty avoidance index), and various structural factors that retard antibiotic stewardship campaigns [46,47,48,49]. Consumption is considerably higher in MSM attending our STI clinics than the general population [50]. One of the drivers of this is intensive screening and treatment of asymptomatic MSM for *N. gonorrhoeae* and *C. trachomatis* which results in macrolide exposures of around 12 defined daily doses per 1000 inhabitants [50]. Considering that this exposure is around 6-fold higher than resistance inducing thresholds in a range of bacteria, it is perhaps not too surprising that commensal *Neisseria* in our study population have acquired a range of mechanisms that enable them to withstand the macrolide selection pressure [51]. Taken in conjunction with the lack of evidence of a benefit of screening for *N. gonorrhoeae* and *C. trachomatis* in MSM, our results provide further evidence to support the reconsideration of this practice [12]. Our results also provide further motivation for the surveillance of AMR in commensal *Neisseria* (and other species) in populations at high risk for the emergence of AMR in *N. gonorrhoeae* and other bacteria [10,15].

## 4. Materials and Methods

### 4.1. Strain Collection and MIC Determination

Between January and May 2019, 10 MSM attending the Institute of Tropical Medicine (ITM) STI clinic with a diagnosis of anogenital gonorrhea were enrolled into this study [15]. After informed consent was obtained, oropharyngeal and anorectal swabs were taken. They were then treated with 500 mg ceftriaxone intramuscularly and 2 g azithromycin orally. The same swabs were taken 14 days later. All swabs were inoculated onto blood and modified Thayer–Martin agar and incubated in 5% carbon dioxide at 36.5 °C for 24 h.

All colonies with a morphology compatible with *Neisseria* were subcultured. Gram staining and oxidase test were performed, and *Neisseria* species were identified by matrix-assisted laser desorption/ionization—time of flight mass spectrometry (MALDI-TOF–MS). Minimum inhibitory concentrations (MICs) of ceftriaxone and azithromycin were determined by E-test. EUCAST breakpoints for *N. gonorrhoeae* were used to define the reduced susceptibilities of all the *Neisseria* species in the panel (azithromycin (AZM) resistant (R) >1 mg/L (ECOFF); ceftriaxone (CRO) resistant (R) ≥0.125 mg/L) [23].

To evaluate the difference in MIC distribution in (i) pathogenic and commensal *Neisseria* and (ii) pre- and post-treatment isolates, a Wilcoxon signed-rank test was performed in R (version 3.6.3) (R Foundation for Statistical Computing, Vienna, Austria).

### 4.2. Whole Genome Sequencing

Genomic DNA was isolated from a single colony of *Neisseria* sps. using MasterPure complete DNA and RNA purification kit (Lucigen Corporation, Middleton, WI, USA) according to the manufacturer’s instructions. Indexed paired-end libraries were prepared using the Nextera XT DNA Library Prep Kit (Illumina, San Diego, CA, USA) and sequenced on an Illumina MiSeq instrument (Illumina, San Diego, CA, USA). Data are available in Genbank: https://www.ncbi.nlm.nih.gov/sra/PRJNA703317 (23 March 2021). Processed Illumina reads were de novo assembled with Shovill (v1.0.4) which uses SPAdes (v3.14.0) using the following: parameters–trim–depth 150–opts–isolate [52,53]. The quality of the contigs were verified with Quast (v5.0.2) [54] followed by annotation using Prokka (v1.14.6) [55].

Additionally, nanopore sequencing was carried for MO0009/1. Genomic DNA was extracted using the MasterPure Complete DNA and RNA Purification Kit (Lucigen Corporation, Middleton, WI, USA), suspended in nuclease-free water and natively barcoded followed by library-preparation (EXP-NBD104, SQK-LSK109). Sequencing was performed on a MinION device with R9.4 flowcell (biosample accession: SAMN18012342). Basecalling was carried out on the fast5 output files using Guppy (Guppy basecalling suite, (C) Oxford Nanopore Technologies, Limited. v3.6.1) to obtain the fastq files. The fastq files were demultiplexed with the guppy barcoder (v3.6.1) followed by Qcat (1.1.0). Porechop (v0.2.4) and filtlong (v0.2.0) were used to trim and filter small reads, respectively [56]. A hybrid assembly was carried out using the trimmed Oxford Nanopore technologies (ONT) and Illumina reads (Trimmomatic v0.39) using Unicycler (v0.4.8), and the assemblies were visually inspected using Bandage (v0.8.1) [57,58,59].

Gene-by-gene analysis was carried out wherein, the core allelic profiles (scheme) of the study strains (n = 26) along with seven reference genome sequences (*N. subflava* (n = 2), *N. gonorrhoeae* (n = 2), *N. meningitidis* (n = 2), *N. mucosa* (n = 1) from National Center for Biotechnology Information (NCBI), were analyzed using (ChewBBACA) [60]. Minimum spanning tree algorithm (MSTree V2) implemented in Grapetree was used to visualize the core genome MLST (cgMLST) allelic loci differences [61].

### 4.3. RAM Analysis and AMR Gene Screening

To extract specific gene sequences (*mtrD, mtrR, penA, porB, ponA, rplD, pilQ*), seqtk (v1.3-r106) was used [62]. The Mtr promoter site was obtained after remapping raw reads on assemblies of the same strain with BWA-MEM (v 0.7.17-r1188) and manually extracted using IGV (v2.8.0) [63,64].

Genes were aligned by using MEGA with the MUltiple Sequence Comparison by Log- Expectation (MUSCLE) algorithm (v10.1.7) [65], except for the hyper variable sequences (*porB* and *mtr* promoter) which were aligned with MAFFT-einsi (v7.471) [66]. SNPs were manually checked based on known RAM positions in *N. gonorrhoeae*.

The protocol from David Eyre et al. was used to determine the number of RAMs of the 23S rRNA alleles [67]. In brief: a bwa index was made of 23S gene sequence (FA1090) to map reads with BWA-MEM followed by sorting and indexing with SAMTOOLS (v1.0) [68]. The ratio of base counts at both RAM positions (A2045 and C2611) were determined with a Python script available at https://github.com/davideyre/gc_mic_prediction_chapter (accessed on 4 March 2020). Abricate (v1.0.1) was used to screen for AMR genes on contigs with default settings (database used: NCBI AMRFinderPlus) [52,69].

### 4.4. Multiple Linear Regression Models of Azithromycin MIC Prediction in Neisseria

Multiple linear regression in R (version 3.6.3) (R Foundation for Statistical Computing, Vienna, Austria) was used to predict the MIC values of azithromycin by using the *lm* function. Log2 transformed values of azithromycin MIC were used as the outcome variable. Known RAMs for azithromycin in *N. gonorrhoeae*/*N. meningitidis* in *mtrD, mtrRP, ponA rplD* were used as predictors in the models. Two models were constructed—one excluding and one including a binary variable representing the absence or presence of *msrD* in the isolate, respectively. The adjusted R squared of both models were compared.

Model 0: MIC_AZMlog2 ~ mtrD_K823E + mtrD_K823S + mtrR_A39T + mtrRP_N.m_like + mtrRP_WHO_P_like + ponA_L421P + rplD_G70S + rplD_G70A

Model 1: MIC_AZMlog2 ~ mtrD_K823E + mtrD_K823S + mtrR_A39T + mtrRP_N.m_like + mtrRP_WHO_P_like + ponA_L421P + rplD_G70S + rplD_G70A + msrD

## Figures and Tables

**Figure 1 pathogens-10-00384-f001:**
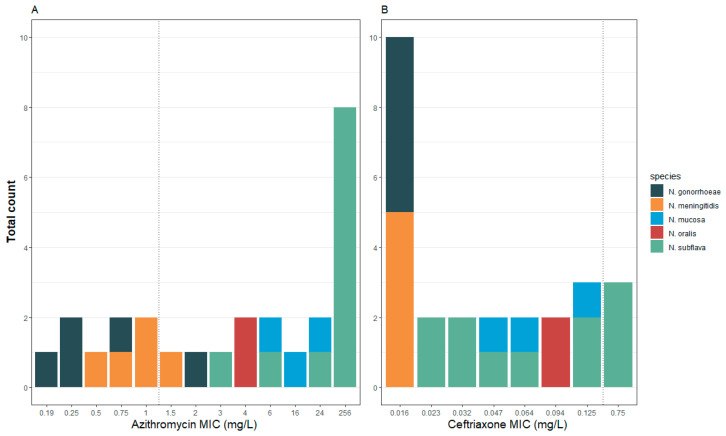
Distribution of the MICs (mg/L) of azithromycin (**A**) and ceftriaxone (**B**) by species in the complete panel of 26 isolates. Dotted line indicates EUCAST breakpoint for *N. gonorrhoeae* (epidemiological cut-off values (ECOFF) for azithromycin) and species are coded by color.

**Figure 2 pathogens-10-00384-f002:**
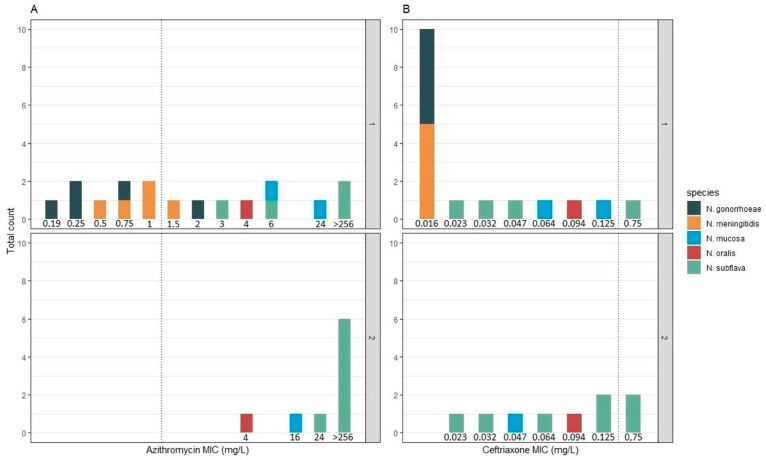
Visit 1 (1) and visit 2 (2) distribution of MICs (mg/L) of azithromycin and ceftriaxone in the panel: Azithromycin MIC distribution from visit 1 and visit 2 (**A**). Ceftriaxone MIC distribution from visit 1 and visit 2 (**B**). Dotted line indicates a breakpoint for N. gonorrhoeae (ECOFF for azithromycin) and species are coded by color.

**Figure 3 pathogens-10-00384-f003:**
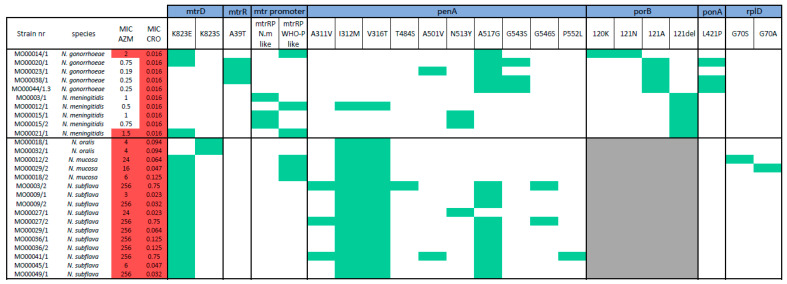
Presence (green) of resistance-associated mutations (RAMs) for macrolide and cephalosporin in this panel. Strains resistant to azithromycin and ceftriaxone are indicated with a red background, grey background indicates hypervariable region at the RAM position compared to pathogenic strains.

**Figure 4 pathogens-10-00384-f004:**
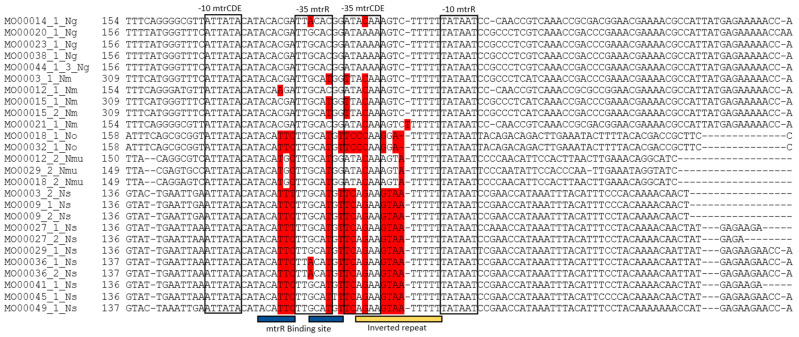
Fragment of alignment of the *mtr* promoter region with *mtrR* and *mtrCDE* promoter sites (black box), *mtrR* binding site (blue bar) and inverted repeat (yellow bar). SNPs in the promoter and binding site compared to the *N. gonorrhoeae* wild type are colored in red. (Strain number with species; *N. gonorrhoeae* (Ng), *N. meningitidis* (Nm), *N. oralis* (No), *N. mucosa* (Nmu) and *N. subflava* (Ns)).

**Figure 5 pathogens-10-00384-f005:**
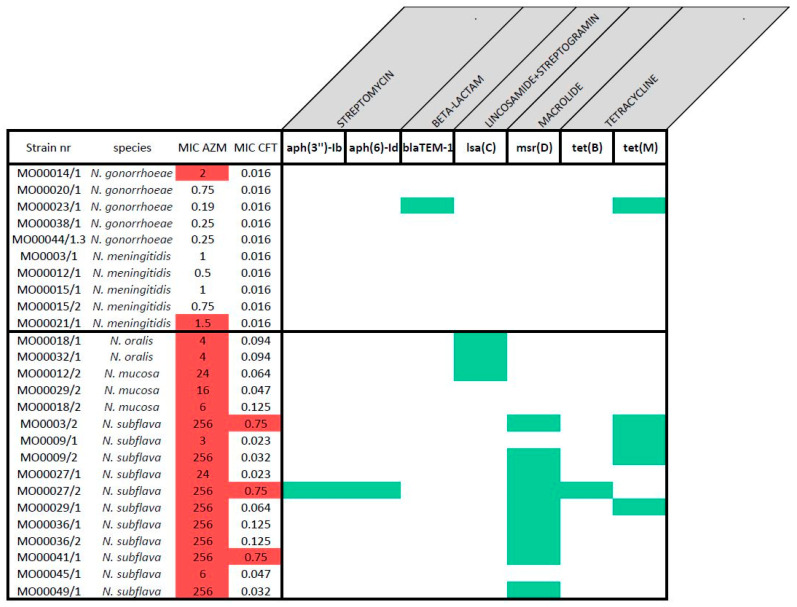
Acquired antimicrobial resistance (AMR) genes (green) in panel. Strains resistant to azithromycin and ceftriaxone are indicated with a red background.

**Figure 6 pathogens-10-00384-f006:**
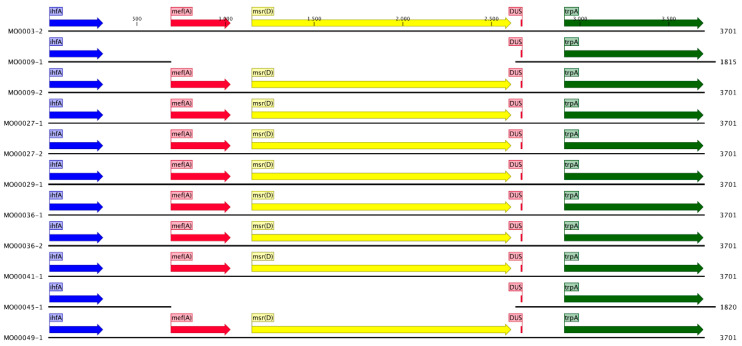
Schematic representation of genomic organization around the integration site. Integration site 30 bp downstream from DNA uptake sequence (DUS) was conserved in all *N. subflava* strains which acquired a portion of the macrolide efflux genetic assembly (MEGA) element. Whilst the *msr(D)* (yellow arrow) gene is full length, only a truncated 338 bp version of the *mef(A)* (red arrow) gene is present.

**Figure 7 pathogens-10-00384-f007:**
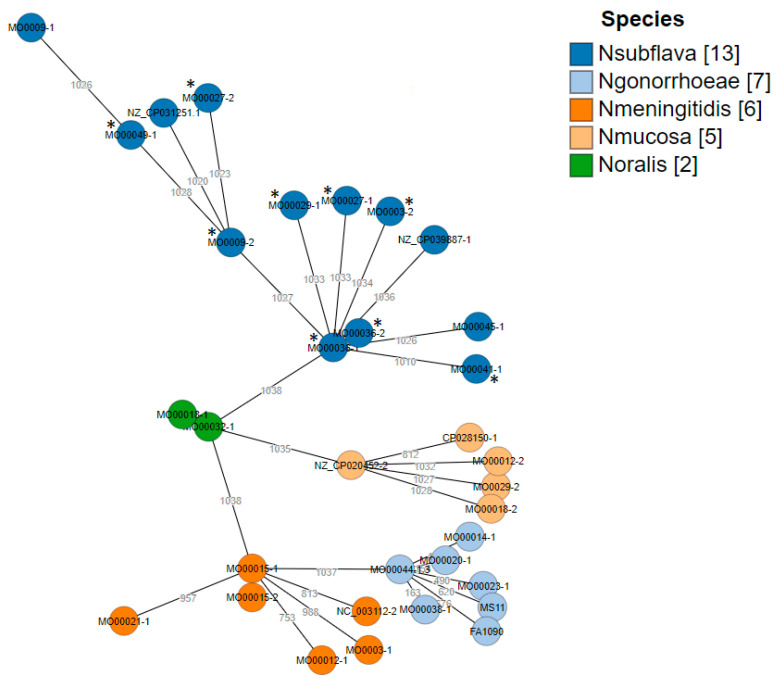
cgMLST hierarchical gene-by-gene analysis of all strains in the panel (n = 26) with reference strains (*N. mucosa*: NZ_CP020452.2/CP020452.2 and CP028150.1. *N. meningitidis:* MC58 and NC_003112.2. *N. gonorrhoeae*: FA1090 and MS11. *N. subflava*: NZ_CP039887.1 and NZ_CP031251.1) based on core genomes. Species are coded by color and asterisks indicate *N. subflava* isolates with *msr(D).*

**Table 1 pathogens-10-00384-t001:** Characterizations of all the strains in the panel.

Species	Strain nr	Visit ^1^	Sampling site ^2^	MIC AZM ^3^	AZM S/R ^4^	MIC CRO ^3^	CRO S/R ^4^
*N. gonorrhoeae*	MO00014/1	1	Anal	2	R	<0.016	S
*N. gonorrhoeae*	MO00020/1	1	Anal	0.75	S	<0.016	S
*N. gonorrhoeae*	MO00023/1	1	Anal	0.19	S	0.016	S
*N. gonorrhoeae*	MO00038/1	1	Anal	0.25	S	0.016	S
*N. gonorrhoeae*	MO00044/1.3	1	Anal	0.25	S	0.016	S
*N. meningitidis*	MO0003/1	1	Oral	1	S	<0.016	S
*N. meningitidis*	MO00012/1	1	Oral	0.5	S	<0.016	S
*N. meningitidis*	MO00015/1	1	Oral	1	S	<0.016	S
*N. meningitidis*	MO00015/2	1	Oral	0.75	S	<0.016	S
*N. meningitidis*	MO00021/1	1	Oral	1.5	R	<0.016	S
*N. oralis*	MO00018/1	1	Oral	4	R	0.094	S
*N. oralis*	MO00032/1	2	Oral	4	R	0.094	S
*N. mucosa*	MO00012/2	1	Oral	24	R	0.064	S
*N. mucosa*	MO00029/2	2	Oral	16	R	0.047	S
*N. mucosa*	MO00018/2	1	Oral	6	R	0.125	S
*N. subflava*	MO0003/2	1	Oral	>256	R	0.75	R
*N. subflava*	MO0009/1	1	Oral	3	R	0.023	S
*N. subflava*	MO0009/2	1	Oral	>256	R	0.032	S
*N. subflava*	MO00027/1	2	Oral	24	R	0.023	S
*N. subflava*	MO00027/2	2	Oral	>256	R	0.75	R
*N. subflava*	MO00029/1	2	Oral	>256	R	0.064	S
*N. subflava*	MO00036/1	2	Oral	>256	R	0.125	S
*N. subflava*	MO00036/2	2	Oral	>256	R	0.125	S
*N. subflava*	MO00041/1	2	Oral	>256	R	0.75	R
*N. subflava*	MO00045/1	1	Oral	6	R	0.047	S
*N. subflava*	MO00049/1	2	Oral	>256	R	0.032	S

^1^ Visit of patient; 1 = day 0, before treatment; 2 = day 14, after treatment. ^2^ Sampling site: oropharyngeal (oral) and anorectal (anal) swabs. ^3^ Minimum inhibitory concentrations of azithromycin (AZM) and ceftriaxone (CRO). ^4^ Breakpoint according to The European Committee on Antimicrobial Susceptibility Testing (EUCAST); azithromycin (AZM) resistant (R) > 1 mg/L; ceftriaxone (CRO) resistant (R) ≥ 0.125 mg/L.

## Data Availability

The data presented in this study are openly available in Genbank at https://www.ncbi.nlm.nih.gov/sra/PRJNA703317, BioProject accession number PRJNA703317.

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
