# Peer review of "WGS of Commensal Neisseria Reveals Acquisition of a New Ribosomal Protection Protein (MsrD) as a Possible Explanation for High Level Azithromycin Resistance in Belgium"

_pathogens, 2021, doi:10.3390/pathogens10030384_

Round 1

Reviewer 1 Report

The paper from de Block and colleagues is a well-written study of the acquisition of a new ribosomal protection protein MrsD as a possible explanation for high level azithromycin resistance in Belgium. This is the first report of this resistance marker in the Neisseriae. I have very few comments to make for this excellent study.

  1. The authors have identified the major limitation of the study that they have not confirmed experimentally that resistance is transferable in vitro. Pity, but I hope that they will do so in the future.
  2. Another limitation that might be worth mentioning is the small cohort of samples. Not a big issue, but should just be highlighted in Discussion.
  3. Minor quibble for acronyms to the non-specialist: page 4, define SNP and RAM.
  4. Again for the non-neisseriologist, why are you looking at porB, mtrCDE and tetM? Provide rationale in Results section when you first mention them.
  5. Figure 7. Core genome. There has been some criticism about FA1090 as a comparator strain; it is somewhat of an outlier in the gonococcal genome maps, so it might be prudent to look at another 'reference' strain, e.g. MS11, etc., just to see what the maximum allelic differences are.
  6. Page 9. last paragraph, there is a comment regarding high antimicrobial consumption in general Belgium population. Can the authors provide a sentence of explanation as to why? Poor stewardship?
  7. Methods section: I think it is always a good idea to provide the MIC determination method (not just ask readers to look at  another paper). Please provide your method (and what reference strain was used).
  8. Methods - WGS. I can see accession numbers in a table, but no information about deposition of sequences anywhere? Have they been added to BIGSdb or Genbank? Can a link be provided?
  9. Ethics approval - please provide the number.
  10. Finally Supplementary Figure 5. Again, for the non-specialist, what am I looking at? Please can you add some description to the legend for this figure. 

Author Response

Comments reviewer 1

  1. The authors have identified the major limitation of the study that they have not confirmed experimentally that resistance is transferable in vitro. Pity, but I hope that they will do so in the future.

Reply: We thank the reviewer for their opinion. It is indeed a pity that we have not confirmed this in vitro, but we are planning to do so in the near future.

  1. Another limitation that might be worth mentioning is the small cohort of samples. Not a big issue, but should just be highlighted in Discussion.

Reply: Indeed this is a limitation. We have adapted this in line 221, mentioning this is a small study:

“In this small study, we explored the genetic determinants of the high azithromycin and ceftriaxone MICs identified in Neisseria isolates from an exploratory study among Belgian MSM.”

  1. Minor quibble for acronyms to the non-specialist: page 4, define SNP and RAM.

Reply: We thank the reviewer for pointing this out. We modified the text on page 4 to define SNP and RAMs.

“…we found 14 single nucleotide polymorphism (SNPs) in commensal Neisseria, which are well established resistance-associated mutations (RAMs) to macrolides/cephalosporins…”

  1. Again for the non-neisseriologist, why are you looking at porB, mtrCDE and tetM? Provide rationale in Results section when you first mention them.

Reply: Indeed, this would help non-neisseriologists. We adapted the text on line 116-117 to explain the common genes related to macrolide and cephalosporins, since we had the focus on these antimicrobials. For tetM, which is an outliner here, we explain this in the discussion (256-261).

Adapted text on line 116-117:

“We aligning genes that are known to mediate macrolide and cephalosporin resistance in N. gonorrhoeae (mtrD, mtrR, penA, ponA, rplD, pilQ and 23S rRNA).”

  1. Figure 7. Core genome. There has been some criticism about FA1090 as a comparator strain; it is somewhat of an outlier in the gonococcal genome maps, so it might be prudent to look at another 'reference' strain, e.g. MS11, etc., just to see what the maximum allelic differences are.

Reply: We thank the reviewer for this comment. Indeed we have also used ‘MS11’ as one of the N. gonorrhoeae reference strain, which is mentioned in line 215 and also represented in Figure 7. We have adapted the text accordingly in lines 210 & 211:

“Maximum allelic differences between N. gonorrhoeae reference strains and the N. subflava core genome were > 3000.“

  1. Page 9. last paragraph, there is a comment regarding high antimicrobial consumption in general Belgium population. Can the authors provide a sentence of explanation as to why? Poor stewardship?

Reply: The reasons for this are complex. We have added the following sentence to provide a broad overview of the likely explanatory factors (272-275):

“This has been linked to a combination of cultural factors (such as a high uncertainty avoidance index), and various structural factors that retard antibiotic stewardship campaigns.”

The four new references we add at the end of this sentence provide detailed analyses of the causes. Deschepper et al., 2002, is a particularly insightful quantative/qualitative analysis that compares responses to typical winter upper respiratory tract type symptoms in the inhabitants of a Dutch and Belgian town 50km apart (and both very close to where we practice). Whereas the Dutch typically attributed their symptoms to a simple viral ‘cold’ and were unlikely to go to a doctor/get antibiotics, the Belgians were more likely to be concerned about bacterial infections and go to a doctor. Here they were frequently diagnosed with ‘bronchitis’ and treated with an antibiotic. The other 3 references unpack the mix of underlying forces that have led to the dramatic difference in antibiotic consumption between these two adjoining countries. 

  1. Methods section: I think it is always a good idea to provide the MIC determination method (not just ask readers to look at another paper). Please provide your method (and what reference strain was used).

Reply: We agree with the reviewer, it is much more convenient to include the compete method. We changed this section accordingly (lines 290-301).

“Between January and May 2019, 10 MSM attending the Institute of Tropical Medicine (ITM) STI clinic with a diagnosis of anogenital gonorrhea were enrolled into this study. After informed consent was obtained, oropharyngeal and anorectal swabs were taken. They were then treated with ceftriaxone 500 mg intramuscularly and azithromycin 2 g orally. The same swabs were taken 14 days later. All swabs were inoculated onto blood and modified Thayer-Martin agar and incubated in 5% carbon dioxide at 36.5°C for 24 hours.

All colonies with a morphology compatible with Neisseria were subcultured. Gram staining and oxidase test were performed, and Neisseria species were identified by matrix-assisted laser desorption/ionization–time of flight mass spectrometry (MALDI-TOF MS). Minimum inhibitory concentrations (MICs) of ceftriaxone and azithromycin were determined by E-test.”

  1. Methods - WGS. I can see accession numbers in a table, but no information about deposition of sequences anywhere? Have they been added to BIGSdb or Genbank? Can a link be provided?

Reply: We thank the reviewer for this comment. Sequences have been added to Genbank. We adapted this in the text in lines 312-313. Data will be released based on publication date of this manuscript.

Adapted text line 312-312:

“Data is available on Genbank: https://www.ncbi.nlm.nih.gov/sra/PRJNA703317.”

  1. Ethics approval - please provide the number.

Reply: In the manuscript format of pathogens, ethics approval is outlined in a separate section. The number of the ethical approval can be found in line 377.

  1. Finally Supplementary Figure 5. Again, for the non-specialist, what am I looking at? Please can you add some description to the legend for this figure.

Reply: We agree with this comment and have updated the legend in this figure.

Reviewer 2 Report

This manuscript expands on an earlier study which characterised the MICs of a panel of non-pathogenic Neisseria isolated from anorectal and oropharyngeal swabs from a cohort of ten Belgian MSMs diagnosed with anorectal gonorrhoea. In that study, increased MICs were noted across all commensal Neisseria strains relative to historic data. In this study a subset of isolates from this cohort, incorporating both pathogenic and non-pathogenic Neisseria, were whole genome sequenced and profiled for the presence of resistance genes, focusing on those linked to macrolide and extended spectrum cephalosporin resistance. The N. gonorrhoeae isolates were from anorectal samples, whereas the remaining isolates originated from oropharyngeal swabs.

The results of the study are interesting and reinforce prior work that suggests commensal Neisseria are an important repository of resistance genes for N. gonorrhoeae. Their data, while from a small cohort of individuals, reinforce the potential utility of WGS-based surveillance of commensal Neisseria.

I do not have a background in WGS, but the methods and approach appear to be appropriate.

The study could have been strengthened by the inclusion of oral isolates of N. gonorrhoeae, given the premise that commensal Neisseria from the oral mucosa are a source of resistance genes. Or better still, profiling both oral and anorectal isolates from the same individual(s), may have been informative. Could the authors comment on this aspect of the study?

Supplementary Files: they are fine except for Supplemental Figure 1, which lacks a figure legend; spelling of pathogen on the x axis of the plot needs to be corrected; and antibiotic name written in full on the y axis.

Author Response

Comment reviewer 2:

This manuscript expands on an earlier study which characterised the MICs of a panel of non-pathogenic Neisseria isolated from anorectal and oropharyngeal swabs from a cohort of ten Belgian MSMs diagnosed with anorectal gonorrhoea. In that study, increased MICs were noted across all commensal Neisseria strains relative to historic data. In this study a subset of isolates from this cohort, incorporating both pathogenic and non-pathogenic Neisseria, were whole genome sequenced and profiled for the presence of resistance genes, focusing on those linked to macrolide and extended spectrum cephalosporin resistance. The N. gonorrhoeae isolates were from anorectal samples, whereas the remaining isolates originated from oropharyngeal swabs.

The results of the study are interesting and reinforce prior work that suggests commensal Neisseria are an important repository of resistance genes for N. gonorrhoeae. Their data, while from a small cohort of individuals, reinforce the potential utility of WGS-based surveillance of commensal Neisseria.

I do not have a background in WGS, but the methods and approach appear to be appropriate.

The study could have been strengthened by the inclusion of oral isolates of N. gonorrhoeae, given the premise that commensal Neisseria from the oral mucosa are a source of resistance genes. Or better still, profiling both oral and anorectal isolates from the same individual(s), may have been informative. Could the authors comment on this aspect of the study?

Reply: We thank the reviewer for their opinion on the manuscript and we fully agree with this comment. In this clinical study we did however not detect any isolates of oral N. gonorrhoeae. Indeed our study would have been strengthened if we had found isolates of oral N. gonorrhoeae. To address this important research question we have boosted our clinical collection of oral N. gonorrhoeae by systematically culturing all PCR positive oral specimens. We plan to do WGS on these isolates and see if there is evidence of horizontal gene transfer between local oral N. gonorrhoeae and commensal Neisseria.

I was unable to open the Supplementary Files, so cannot comment on them.

Supplementary Files: they are fine except for Supplemental Figure 1, which lacks a figure legend; spelling of pathogen on the x axis of the plot needs to be corrected; and antibiotic name written in full on the y axis.

Reply: We thank the reviewer for this observation. We corrected the figure accordingly.